# Effects of Playing Exergames on Quality of Life among Young Adults: A 12-Week Randomized Controlled Trial

**DOI:** 10.3390/ijerph20021359

**Published:** 2023-01-11

**Authors:** Jiajun Yu, Han-Chung Huang, T. C. E. Cheng, May-Kuen Wong, Ching-I Teng

**Affiliations:** 1School of Management, Guangzhou Huashang College, Guangzhou 511300, China; 2School of Innovation and Entrepreneurship, Guangzhou Huashang College, Guangzhou 511300, China; 3Graduate Institute of Management, Chang Gung University, Taoyuan 333, Taiwan; 4Center for General Education, China University of Technology, Taipei 219, Taiwan; 5Department of Logistics and Maritime Studies, The Hong Kong Polytechnic University, Hong Kong 999077, China; 6Taoyuan Branch, Chang Gung Memorial Hospital, Taoyuan 333, Taiwan; 7Department of Physical Medicine and Rehabilitation, Linkou Chang Gung Memorial Hospital, Taoyuan 333, Taiwan; 8Department of Business and Management, Ming Chi University of Technology, New Taipei 243, Taiwan

**Keywords:** exergames, active games, randomized controlled trial, quality of life, SF36, health

## Abstract

**Objective:** The purpose of this paper is to investigate whether playing exergames can enhance quality of life among young adults and it examines the potential moderators. **Methods:** A 12-week randomized controlled trial was conducted. Quality of life was measured using the short-form 36-item version (SF-36) scale. All the participants were between 20 and 24 years old in Taiwan. Participants in the intervention group (*n* = 55) were asked to play exergames for 12 weeks, three times a week and 30 minutes at a time, while participants in the control group (*n* = 62) did not play exergames. The changes in the scores on quality of life between the beginning and the end of the 12-week trial were calculated. Independent *t*-tests were used to analyze the differences. **Results:** The intervention group participants experienced an enhanced quality of life in terms of physical functioning, role-physical (role limitations due to physical health), general health, and social functioning. Moreover, the intervention group participants who were not enthusiastic about exercisers experienced an enhanced quality of life in physical functioning, role-physical, and general health. The intervention group participants who attempted to control their weight experienced enhanced general health, vitality, and mental health. **Conclusion:** Playing exergaming could contribute to users’ quality of life in terms of both physical and mental health.

## 1. Introduction

Exergames are videogames that translate body motions into gaming commands, and they could be used to enhance users’ health. Exergaming is a new emerging form of exercise that could improve physical activity and psychosocial well-being [1,2]. Exergames are prevalent worldwide, as evidenced by the fact that Your Shape Fitness Evolved 2012, an exergame, has sold 1.16 million units [3]. The prevalence of exergames warrants further research on the impact of playing exergames on users’ health.

The literature has documented various health benefits of playing exergames to individuals’ health [4,5]. However, previous studies have not examined whether playing exergames can promote quality of life, one widely recognized aspect of health, indicating a research gap.

Research filling this gap could ascertain the impact of playing exergames on players’ health in terms of quality of life. Moreover, such research could provide evidence for the health impact of playing exergames on users’ life, further demonstrating the value of games to users’ health.

Research on quality of life has identified the importance of exercise. Exercise could promote quality of life [6]. Engaging in exercise can be accomplished by playing exergames, so the impact of playing exergames should be related to how individuals are strongly motivated to do exercise, i.e., exercise enthusiasm. However, no study has examined the moderator role of exercise enthusiasm in the link between playing exergames and quality of life, indicating another research gap.

Research filling this gap can extend our knowledge on the health benefits of playing exergames, i.e., clarifying who (e.g., highly or lowly enthusiastic about doing exercise) would receive enhanced benefits to quality of life by playing exergames. In practice, such research may provide insights for healthcare professionals to enhance the quality of life of their clients. In sum, such research exhibits academic significance and practical relevance.

Seeking to fill the above two research gaps, we design and implement this research to examine whether playing exergames can enhance quality of life and examine some potential moderators. Exercise enthusiasm and weight control were included as two moderators for the following reasons. Enthusiasm (or passion) could create positive experience in sports [7], so it is relevant to research on quality of life, justifying its inclusion. Moreover, weight control is related to physical functioning and self-esteem, so it could be related to changes in quality of life [8], justifying its inclusion.

Compared with previous studies, our research has two contributions to the literature. First, Lee et al. [9] found that exergames could generate acute effects to increase users’ physical activity which might reduce a negative mood state. In line with Lee et al. [9], the impact of playing exergames were also examined herein, but a new beginning was conducted to examine the positive impact of playing exergames on quality of life, motivating subsequent studies to examine the mechanism underlying the impact of playing exergames, as well as the accompanying enjoyment on quality of life.

Second, Ho, Lwin, Sng, and Yee [10] found that playing exergames could create a feeling of presence, thus creating a positive psychological experience and enjoyment. Consistent with Ho et al. [10], the psychological impact of playing exergames on users’ psychological well-being were also examined. However, a unique case herein was shown in examining quality of life, a broadly used measure of well-being, supporting the relevance of playing exergames to enhancing psychological well-being [11].

The rest of the paper is organized as follows. We firstly review the literature and develop the study hypotheses. We secondly describe the methods and the testing results. We then summarize the research findings, and discuss their theoretical and practical implications. Finally, limitations and conclusions are drawn.

## 2. Literature Review

### 2.1. Exergames

Exergames are video games that combine entertainment and body movement on a video device such as a TV screen or computer monitor. Exergames are useful tools to increase exercise capacity in patients with chronic heart failure [5]. Moreover, exergames have exhibited various influences on individuals’ health [12,13]. Specifically, playing exergames results in energy expenditure, so creating positive psychological benefits [14]. Moreover, playing exergames significantly improves users’ attitudes towards other forms of exercise [15] and motivates users to engage in exercise, so exhibiting health benefits [16], which shows the value of exergames.

In addition, playing exergames may have a similar effect to engaging in conventional exercise. Specifically, exergaming can promote engagement in physical activities [9,17]. Playing exergames can reduce pressure and pain sensitivity [12], and help motivate prosocial behaviors [18]. Moreover, playing exergames can effectively instill a positive mood [19] and trigger health-related behavioral changes [20]. Hence, playing exergames was considered as a form of exercise to improve health outcomes.

Playing exergames can promote health behavior change [9,20], and facilitate positive psychosocial outcomes, identifying the key elements of exergaming that enable social interaction [21]. Playing exergames can also create a feeling of presence, enjoyment, and mood experience [10], generating a greater amount and diversity of reward impacting on the users’ experience [22,23]. The literature has found that exergaming plays a basic role in producing positive effects, positive functioning, and positive social functioning, contributing to mental health and wellness [24]. Hence, exergames can be used as an alternative strategy tool for health interventions.

Playing exergames can promote engagement and motivation to do exercise, so bringing positive health benefits [16]. The literature has employed the decomposition techniques of the Physical Component Score (PCS) and Mental Component Score (MCS) from the SF-36 scale as measures of health [25]. However, quality of life comprises eight aspects that should further be examined in relation to exergaming, indicating fruitful ground for research opportunities in the future.

### 2.2. Quality of Life and SF-36

Comprising the physical, emotional, and social dimensions, quality of life is suitable to represent the multi-dimensional aspects of health status [26]. Among the measures of quality of life, the SF-36 instrument is frequently used in health assessment [25]. Specifically, SF-36 consists of eight dimensions of health, namely, physical functioning, role-physical (role limitation due to physical problems), bodily pain, general health, vitality, social functioning, role-emotional (role limitation due to emotional problems), and mental health.

## 3. Hypothesis Development

Playing exergames can motivate users’ engagement in physical activities that resemble exercise [1,13]. Thus, playing exergames can be employed as an effective strategy for motivating users to engage in exercise [16]. In addition, playing exergames improves users’ physical fitness [27] and mood state [28], which contribute to their physical and psychological health, building the link between playing exergames and enhanced quality of life. Moreover, playing exergames can increase users’ physical activity levels [13], while physical activities promote sleep quality [29], which is essential for ensuring quality of life, supporting the impact of playing exergames on enhanced quality of life. Hence, the following was hypothesized:

**H1.** 
*Playing exergames can enhance the quality of life of the participants in the intervention group more than those in the control group.*


Exercise enthusiasm positively predicted positive emotions in exercise [30]. Playing exergames has a positive impact on users’ attitude and intention towards other forms of exercise [15], indicating that playing exergames may be a means for enhancing exercise enthusiasm [31]. Exercise enthusiasm should provide additional enjoyment in doing exercise, so effectively improving quality of life and physical health [32]. Therefore, playing exergames is more likely to enhance the quality of life of individuals’ who are enthusiastic about exercise. Hence, the following was hypothesized:

**H2.** 
*Exercise enthusiasm is more positively associated with the quality of life of the participants in the intervention group than those in the control group.*


Weight control is strongly related to quality of life [8], the reason for which may be due to the link between obesity and reduced quality of life [33]. Quality of life plays a critical role for the overall assessment of being overweight and obesity [34]. Being overweight and obesity have a negative impact on quality of life [35]. Weight control can be achieved by dieting or exercise [36]. Exercise is embedded in playing exergames, so playing exergames should enhance the quality of life of individuals’ who are controlling their weight. Hence, the following was hypothesized:

**H3.** 
*Weight control is more positively associated with the quality of life of the participants in the intervention group than those in the control group.*


## 4. Methods

### 4.1. Design, Experiment, and Participants

Prior to carrying out the study, approval by the Institutional Review Board (102-3841B) was obtained. This research was conducted in a spacious classroom at a university in north Taiwan, between January 2014 and April 2014. The study was a parallel-group randomized controlled trial with 117 participants who were undergraduate and postgraduate students, ranging in age from 20 to 24 years old. The participants were solicited by hardcopy posters in the campus. Interested participants completed an online form to offer their contact information for further checking their eligibility of joining this research. The enrolled participants were randomly assigned to the intervention group and the control group using a random number table.

Eligible participants were those who could communicate in Chinese, effectively taking part in the study and answering the questionnaire at the beginning and the end of research participation. Each participant offered written informed consent to join the study. The participants who reported the following health conditions were excluded: heart disease, hypertension, arrhythmia, heart failure, angina pectoris, spinal cord injury, ankylosing spondylitis, genetic arrhythmia, intracranial pressure, blood pressure instability, cervical musculoskeletal system injury, glaucoma, high myopia, anemia, dizziness, asthma, or had any other mental illness or disease history.

A 12-week longitudinal design was adopted. The intervention group participants (*n* = 55) underwent a 12-week period of exergames, playing 30 minutes per game and three times a week. The control group participants (*n* = 62) did not play exergames during the trial period. In terms of hardware, Microsoft’s Xbox 360 was used, which has a control-free system allowing gamers to use their body to control gaming avatars. As for the exergame, the “Your Shape: Fitness Evolved” program was adopted, which provides a wide choice of activities to track subtle movements of the body.

The participants were asked to actually play exergames during the 12 weeks. The research assistants made appointments with them and supervised them to actually play the exergames. To maintain the participants’ patients, ten exergames were randomized for the participants to play. This approach increases the generality of the study findings.

### 4.2. Measurement

The formula of Schoenfeld [37] was used to calculate the required sample size for a randomized controlled trial. A parallel experiment design was used and continuous dependent variables with the one-tailed significance level set at 5% and the testing power set at 80% were configured. The estimated sample size is 113. Totally, 117 participants were included, exceeding the threshold.

The participants were randomly assigned to one of the two groups to maintain the internal validity of the randomized trial. All the participants were asked to sign an informed consent form. The participants were invited to complete the questionnaire at the beginning and end of the study.

The primary outcome variables of our research were mental and physical health. The participants’ mental and physical health were measured using the SF-36 scale. This scale contains various dimensions of quality of life. The following describes the number of items for each dimension and an example item. The dimensions are: physical functioning (10 items, e.g., “Does your health hinder you from doing effortful daily activities, e.g., running? (from “not at all” to “very much”)”); role-physical (role limitation due to physical problems) (four items, e.g., “During the past four weeks, have your health caused a problem in doing your work? (“no” or “yes”)”); bodily pain (two items, e.g., “During the past four weeks, how severe your bodily pain is? (from “not at all” to “very severe”)”); general health (five items, e.g., “Generally, how do you think your health is? (from “not good” to “perfect”)”); vitality (four items, e.g., “During the past four weeks, how often do you feel energetic? (from “never” to “always”)”); social functioning (two items, e.g., “During the past four weeks, how much does your health hinder your activities with family, friends, neighbors, and clubs? (from “not at all” to “extremely”)”); role-emotional (role limitation due to emotional problems) (three items, e.g., “During the past four weeks, have your emotional problems (e.g., anxiety or depression) reduced your work time? (“no” or “yes”)”); and mental health (six items, e.g., “During the past four weeks, how often are you a nervous person? (from “never” to “always”)”). The items measuring each dimension contain varied response options. Please consult SF-36 instrument owners to check how to calculate the scores in each of the dimensions.

Each of the participants was asked two questions: “whether or not you are an enthusiastic exerciser”. If the answer was “yes”, we coded this response as “1”. If the answer was “no”, we coded this response as “0”, and “whether or not you are controlling your weight”. If the answer as “yes”, we coded this response as “1”. If the answer was “no”, we coded this response as “0”. The research assistants distributed and collected the questionnaires. Moreover, this research was designed to ensure that all research assistants were blind to the participant assignments. The exceptions were the data managers who did not directly contact the participants and did not operate the experiment.

### 4.3. Data Analysis

After recording the original dimension scores, these scores were converted from zero to one hundred [25]. A higher converted score represents a higher level in a dimension. Independent sample *t*-tests were used to examine whether playing exergames enhances quality of life. Moreover, the responses by subsamples were compared using the *t*-tests for testing the moderator hypotheses. The SPSS software version 22 (IBM, Armonk, New York, USA) was employed to analyze the data. The significance level at 0.05 and the marginal significance level was set at 0.10.

## 5. Results

Cronbach’s α values were used to evaluate the internal consistency of the scale. In the SF-36 scale, each construct measure has a Cronbach’s α value approaching 0.70, indicating sufficient reliability [38]. The items evaluating each construct have a Cronbach’s α > 0.70, except for social functioning, which equals 0.65.

The marginal change amounts of physical functioning (*p* = 0.03), role-physical (*p* = 0.03), general health (*p* = 0.01), and social functioning (*p* = 0.07) for the intervention group were significantly better than the control group. These results supported H1. However, the marginal change for the intervention group was not significantly better than the control group in other aspects of quality of life. Table 1 reported the results.

To keep clarity, only the statistics for significant results for the moderating effects were reported. The marginal change amount of the intervention group of participants that have no exercise enthusiasm is significantly better than the control group in general health (8.81 vs. 0.00, *p* = 0.02). The marginal change amount of the intervention group of participants that had exercise enthusiasm was significantly better than the control group in physical functioning (3.13 vs. −0.51, *p* = 0.02), role-physical (7.81 vs. −3.21, *p* = 0.09), and general health (4.50 vs. −1.05, *p* = 0.05). These resulted support H2.

The marginal change amount of the intervention group of participants that were controlling weight was significantly better than the control group in physical functioning (3.04 vs. −0.54, *p* = 0.05) and general health (5.00 vs. −0.73, *p* = 0.02). The marginal change amount of the intervention group of participants that were controlling weight was significantly better than the control group in general health (6.80 vs. −0.70, *p* = 0.06), vitality (2.00 vs. −7.75, *p* = 0.01), and mental health (−0.48 vs. −4.80, *p* = 0.07), supporting H3.

## 6. Discussion

### 6.1. Main Findings

The analytical results indicate that playing exergames could enhance quality of life, in terms of physical functioning, role-physical, general health, and social functioning. In other words, playing exergames could effectively enhance quality of life, mostly in terms of physical health. In addition, playing exergames might provide enhanced benefits in quality of life among individuals who were enthusiastic about exercise or attempting to control their weight.

### 6.2. Theoretical Implications

Our research identified a novel impact of playing exergames, i.e., playing exergames is positive for quality of life. This research thus answered Li and Lwin’s [39] call for more research on the impact of playing exergames. By answering this call for research, our findings deepen our understanding of the benefits of playing exergames, so motivating future works in this line of research.

Our research examined the relationship between playing exergames and quality of life, in terms of physical functioning, role-physical, general health, and social functioning. These findings are consistent with the findings of Maitland et al. [21] on the link between exergames and psychosocial outcomes. However, our research attempt was unique in exploring the relationship between exergames and quality of life among healthy young adults, one significant sector of the population that has an increasing need for more exercise and social interaction.

Our study adopted a longitudinal design and, by using this design, examined quality of life issues. This examination is in concordance with the study of Peña and Kim [23], who found that playing exergames affected users’ physical activity, and so encouraged healthier outcomes. However, our research uniquely examined whether playing exergames might improve quality of life. The exergaming literature indicates that playing exergames could improve users’ agility, body coordination [40], body balance, postural stability [40], and mental health and wellness [24]. Therefore, the research findings herein supported that exergames should be further promoted to the wider community to enhance people’s health and quality of life.

Our research employed a randomized controlled trial to obtain evidence that young adults can have positive quality-of-life outcomes. This research is in line with the study of Johnson et al. [22], who found that playing exergames generated a greater amount and diversity of rewards impacting on the users’ experience. Moreover, our findings uniquely indicate that playing exergames could enhance quality of life for users, providing users with valuable information on which to base purchasing decisions.

Recent exergaming literature has examined the impact of playing exergames on the intention to engage in other forms of exercise [15], moods [28], and physical fitness [27], while Huang et al. [41] found that flow experience could promote users to repeatedly play exergames, thus benefiting their health. Compared with those studies, this study is new in examining the impact of playing exergames on users’ quality of life, a novel consequence of playing exergames, exhibiting the newness of our research work.

### 6.3. Implications for Practitioners

Playing exergames was observed as being able to improve quality of life in practice. Therefore, exergames may be used in promotional and educational activities that aim to engage the public in health-promoting activities and improve their quality of life. On the other hand, more persuasive evidence of the health benefits of playing exergames were provided for managers to promote their exergames.

Our analytical results demonstrate that playing exergames could enhance individuals’ physical and mental health. Therefore, users could actively play exergames for improving their physical and mental health. Moreover, exergame providers could hold exergaming contests or incorporate exergames into their health promotion programs. Such incorporation may enhance the impact of exergames on the population’s health. Furthermore, healthcare professionals may encourage their clients to play exergames, so as to enhance their clients’ quality of life.

According to our study findings, playing exergames may enhance quality of life in terms of the mental health of players who are enthusiastic about exercise or attempting to control their weight. Accordingly, it can be suggested that individuals fitting such conditions try to play exergames to reap the enhanced health benefits of playing exergames, in terms of enhancing their quality of life. Moreover, we recommend that playing exergames, as an effective strategy, can also serve public health in the community.

### 6.4. Research Limitations and Future Research Directions

We recruited the participants who were all college students. These participants were suitable because of their homogeneity and were unlikely to introduce features that would bias the research results. To increase the generality of our research findings, future studies could replicate the research work by recruiting participants other than students.

The influence of playing exergames on users’ physical and psychological health were explored, but emotional impact was not included. Emotions may affect the continual intention of the participants. Therefore, future works might explore whether playing exergames could effectively trigger emotional changes, so boosting continual intention of playing exergames.

A randomized controlled trial was used, one of the most rigorous experimental designs to investigate whether and for whom playing exergames enhances quality of life. However, the mechanism underlying the positive impacts of playing exergames on quality of life was not explored. Therefore, future works might apply qualitative research designs to further uncover the detailed process underlying the findings herein.

## 7. Conclusions

Our study is one of the first studies in examining whether playing exergames improves young adults’ quality of life. This is important, because we do not clearly know whether the introduction of exergames benefits quality of life in this population. Our study advances our understanding by indicating that the answer is yes, but only for certain aspects of quality of life, i.e., in physical functioning, role-physical (role limitations due to physical health), general health, and social functioning. Hence, our findings make contributions to exergame providers and users by informing them: exergames can be useful to those who are insufficient in the above-listed aspects. Playing exergames plays an important role in improving public health [42], treating obesity [43], and rehabilitating patients [44]. Our research establishes a link effect that playing exergames might enhance quality of life, in terms of physical and mental health. Such enhancements were stronger for those who were enthusiastic about exercise or attempting to control their weight, contributing to the understanding of the health effects of playing exergames. However, future works need to explore how and why playing exergames could improve users’ quality of life, offering a comprehensive understanding of the underlying process.

## Figures and Tables

**Table 1 ijerph-20-01359-t001:** Comparison of changes in quality-of-life scores between the two groups.

Variable	Intervention Group			Control Group			*p* Value
	Post	Pre	Diff	Post	Pre	Diff	
Physical functioning	97.08	94.58	2.50	96.05	95.96	0.09	0.03 *
Role-physical	90.10	84.37	5.73	80.70	86.84	−6.14	0.03 *
Bodily pain	86.84	83.15	3.31	84.80	85.40	−0.60	0.19
General health	76.79	70.85	5.94	69.39	70.11	−0.72	0.01 *
Vitality	67.19	65.73	1.46	64.74	65.26	−0.52	0.41
Social functioning	86.98	84.38	2.60	82.46	84.43	−1.97	0.07 *
Role-emotional	80.56	72.92	7.64	77.19	73.10	4.09	0.64
Mental health	70.25	71.42	−1.17	67.36	69.61	−2.25	0.29

*Note.* * denotes significant between-group difference (*p* < 0.05). Post = post-test scores. Pre = pre-test scores. Diff = difference between the two left columns.

## Data Availability

The authors do not have the right to share the data.

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
