# Peer review of "Effects of Playing Exergames on Quality of Life among Young Adults: A 12-Week Randomized Controlled Trial"

_ijerph, 2023, doi:10.3390/ijerph20021359_

Round 1
Reviewer 1 Report
Thank you for the opportunity to review this manuscript which I found to be generally well-written and interesting. The study of exergames and their impact on young adults is indeed interesting.
Abstract
Clear abstract, explaining the methods of the study and its findings.
Introduction
A clear purpose is discussed, along with relevant theoretical grounds. On the last paragraph of the Introduction, at the end, perhaps you can add another sentence, explaining the structure of the paper. This would be beneficial to the reader, who will be able to formulate a clearer understanding in terms of organization and navigation within the text.
Conceptualization
The analysis is detailed, and includes a range of updated sources, published between 2017 and 2022.
Conclusion
In the discussion, perhaps you can a) discuss in more depth the main points of the paper, and b) make a stronger case for the original contributions the paper makes. In other words, why is this study needed now and how does it advance our understanding of relevant theoretical or empirical matters?
Very good work overall. I look forward to receiving a revised version of the paper.
Cordially,
Author Response
Point 1: Thank you for the opportunity to review this manuscript which I found to be generally well-written and interesting. The study of exergames and their impact on young adults is indeed interesting.
Response 1: Thank you for offering the recommendation of minor revision. We greatly appreciate your recommendation and the helpful comments. Thank you for indicating that this paper is generally well-written and interesting. We believe that publication of this paper can further increase the impact of this journal.
Point 2: Abstract. Clear abstract, explaining the methods of the study and its findings.
Response 2: Thank you for indicating that the abstract currently clearly explains the methods and the findings.
Point 3: Introduction. A clear purpose is discussed, along with relevant theoretical grounds. On the last paragraph of the Introduction, at the end, perhaps you can add another sentence, explaining the structure of the paper. This would be beneficial to the reader, who will be able to formulate a clearer understanding in terms of organization and navigation within the text.
Response 3: We greatly appreciate your positive remarks, i.e., there is a clear purpose, relevant theoretical grounds. Thank you for the clear instruction. As instructed, we add one new paragraph to explain the structure of the paper. Yes, we totally agree that the addition of this sentence helps readers navigate through the text of the revised paper.
Point 4: Conceptualization. The analysis is detailed, and includes a range of updated sources, published between 2017 and 2022.
Response 4: Thank you for indicating that this paper has cited papers published between 2017 and 2022, thus enabling connection to the recent literature. We greatly appreciate your approval.
Point 5: Conclusion. In the discussion, perhaps you can a) discuss in more depth the main points of the paper, and b) make a stronger case for the original contributions the paper makes. In other words, why is this study needed now and how does it advance our understanding of relevant theoretical or empirical matters?
Response 5: Thank you for the clear instruction on revising the Conclusion section. During this revision, we discuss the main points of this paper and discuss who this study is needed and how it advances our understanding (the first eight lines of the revised Conclusion section).
Point 6: Very good work overall. I look forward to receiving a revised version of the paper.
Response 6: Thank you for the approval. We greatly appreciate your encouraging remarks, i.e., this work is good overall. We closely follow your advices to improve this paper. Hope the revised version can reach an acceptable level. If not, we are willing to make further amendments to improve it.
Reviewer 2 Report
This could be a useful contribution but would need some considerable revision.
To start with, please delete most or all of the phrase ‘the present study’, which occurs 63 times. It is obvious you are writing about the present study, and you can instead put things in the past tense, for example (line 17) ‘quality of life was measured …’.
Abstract: please mention the ages of participants, and the country (Taiwan).
Introduction: covers a lot of useful references, but the organization can be improved – we get a literature review on pp.1-2 and then a heading Literature review on p.3! so some repetition (for example defining evergames) can be reduced.
Method: bottom p.4, tell us more about how the participants were recruited.
p.5 top line is not very clear. How many games were played? Was this based on self-report or do we have any independent evidence that these games were actually played?
p.5 Measurement – we need a lot more detail about what measures were used! This includes all the main measures in Table 1, plus the measures of exercise enthusiasm, and controlling weight.
Results: Table 1 gives change scores, but we should also be given the full pre-test and post-test scores for both Intervention and Control groups.
Discussion: this should focus more on your findings. At present, you foreground other studies and then say e.g. that you supported it (or not). It is better to foreground your findings, and then relate to other studies as appropriate.
p.8 line 350 you tell us the participants were undergraduate and postgraduate students. That information should be included in Methods.
Author Response
Point 1: This could be a useful contribution but would need some considerable revision.
Response 1: Thank you for offering an opportunity to revise this paper. We implement considerable revision by closely following your helpful suggestions. We believe that these helpful suggestions have greatly improved the quality of this paper. The following explains how each of the suggestion is adopted for improving this paper.
Point 2: To start with, please delete most or all of the phrase ‘the present study’, which occurs 63 times. It is obvious you are writing about the present study, and you can instead put things in the past tense, for example (line 17) ‘quality of life was measured …’.
Response 2: Thank you very much for the clear instruction. As instructed, we totally wipe out all the uses of “the present study” throughout the revised paper. Thank you for the helpful suggestion, i.e., using the past tense. We used a number of times of past tense for removing the words: “the present study”, as possible and as appropriate. Thank you!
Point 3: Abstract: please mention the ages of participants, and the country (Taiwan).
Response 3: Thank you for the clear instruction. As instructed, we explained that the participants were between 20 and 24 years old. Moreover, we also clarified that the participants were in Taiwan. We disclosed the above information in the Abstract section (the fourth line).
Point 4: Introduction: covers a lot of useful references, but the organization can be improved – we get a literature review on pp.1-2 and then a heading Literature review on p.3! so some repetition (for example defining evergames) can be reduced.
Response 4: Thank you for the helpful suggestions. As suggested we reorganized the Introduction section. First, we reduced some repetitive citations, particularly those defining exergames. Second, we removed some citations that were restated in other sections. Third, while keeping useful references, we reduce the page length of the Introduction section (reducing 17 lines), to make the Literature Review section title appearing on page 2, one page earlier than the firstly submitted version.
Point 5: Method: bottom p.4, tell us more about how the participants were recruited.
Response 5: Thank you for the clear instruction. As instruction, we explain more about how the participants were recruited. The information was added to the Methods section (the first paragraph, the lines 5-7).
Point 6: p.5 top line is not very clear. How many games were played? Was this based on self-report or do we have any independent evidence that these games were actually played?
Response 6: Thank you for asking this point. We should have clearly explained this point. The research assistants made appointments with the participants and supervised them to actually play the exergames. Therefore, playing exergames is not self-reported, but actually implemented with independent evidence. We also explained the number of games played and justified our choices. The aforementioned information was added to the revised paper, as a new paragraph immediately above the Measurement subsection title.
Point 7: p.5 Measurement – we need a lot more detail about what measures were used! This includes all the main measures in Table 1, plus the measures of exercise enthusiasm, and controlling weight.
Response 7: We are sorry that SF-36 is a registered intellectual property. We cannot show it in full. Otherwise, we would infringe the legal rights of the SF-36 owners. Thank you for asking. In the firstly submitted version of this paper, the measurement information of exercise enthusiasm and controlling weight were reported but not sufficiently clear. We provided more detail about the measures of exercise enthusiasm and controlling weight in the Measurement subsection, the last paragraph, in its lines 6-8.
Point 8: Results: Table 1 gives change scores, but we should also be given the full pre-test and post-test scores for both Intervention and Control groups.
Response 8: Thank you for indicating that we should prepare fuller information in Table 1, including pre-test and post-test scores for both Intervention and Control groups. As required, we added the values to Table 1.
Point 9: Discussion: this should focus more on your findings. At present, you foreground other studies and then say e.g. that you supported it (or not). It is better to foreground your findings, and then relate to other studies as appropriate.
Response 9: Thank you for providing the exact instruction. As instructed, we reorganize each paragraph of the Theoretical Implications subsection of the Discussion section. Specifically, we foreground our research findings first and then relate the findings to other studies as appropriate.
Point 10: p.8 line 350 you tell us the participants were undergraduate and postgraduate students. That information should be included in Methods.
Response 10: Thank you for the clear instruction. As instructed, such information as moved to the Methods section in its first paragraph, lines 4-5.
Round 2
Reviewer 2 Report
Some corrections have been made, but a couple of issues remain.
p.5 Measurement – we still need more detail about what measures were used!
In particular be precise about the scales in SF-36 employed, the number of items in each scale, and mention one sample item.
Results: Table 1 is not properly explained. Saying ‘All the numbers are changes between the beginning and the end of the study’ is unhelpful (and incorrect). It should be clear which are Pre and which Post scores.
Author Response
Response to Reviewers 2 Comments
Thank you for the further suggestions for improving this paper! We tracked our changes to clarify the revision process.
Point 1: Some corrections have been made, but a couple of issues remain
Response 1: Thank you for offering an opportunity to revise this paper. We feel fortunate that you feel all other corrections are made properly, while a couple of issues remain. We fully address these issues, as listed below. Your insightful advice is highly appreciated for further improving this paper to a higher quality!
Point 2: p.5 Measurement - we still need more detail about what measures were used!
Response 2: Thank you for indicating the need to explain in more details about what measures were used. We believe that below you have a focus on SF-36. So, we use the space here to particularly and firstly explain the two other variables, i.e., exercise enthusiasm and weight control. The two variables and quality of life (SF-36) are everything mentioned in the study hypotheses.
Each of the participants was asked two questions: “whether or not you are an enthusiastic exerciser”. If the answer was “yes”, we coded this response as “1”. If the answer was “no”, we coded this response as “0”, and “whether or not you are controlling your weight”. If the answer as “yes”, we coded this response as “1”. If the answer was “no”, we coded this response as “0”.
We reported the above details in the fourth paragraph of the Measurement subsection.
Point 3: In particular be precise about the scales in SF-36 employed, the number of items in each scale, and mention one sample item.
Response 3: Thank you for indicating the need to explain the SF-36 scale, i.e., the number of items in each scale, and mention one sample item. We exactly followed your instruction to report all the required information (as in the third paragraph of the Measurement subsection).
Point 4: Results: Table 1 is not properly explained. Saying 'All the numbers are changes between the beginning and the end of the study' is unhelpful (and incorrect). It should be clear which are Pre and which Post scores.
Response 4: Thank you for pointing out the incorrect table note. We fully agreed with you and removed the note you mentioned. Instead, we wrote “* denotes significant between-group difference (p < .05). Post=post-test scores. Pre=pre-test scores. Diff=difference between the two left columns.” (the note below Table 1). We also use columns to display the Post scores, Pre scores, and their differences, for each group. Your insightful comment boosts the clarity of Table 1. We greatly appreciate your guidance.